# Research on the Effect of Manufacturing Agglomeration on Green Use Efficiency of Industrial Land

**DOI:** 10.3390/ijerph20021575

**Published:** 2023-01-15

**Authors:** Yuan Wang, Anlu Zhang, Min Min, Ke Zhao, Weiyan Hu, Fude Qin

**Affiliations:** 1College of Public Administration, Huazhong Agriculture University, Wuhan 430070, China; 2Research Center for Natural Resources Management and Global Governance, Huazhong Agricultural University, Wuhan 430070, China; 3Aeronautics and Intelligent Manufacturing College, Hankou University, Wuhan 430212, China

**Keywords:** manufacturing agglomeration, green use efficiency of industrial land, spatial and temporal differentiation characteristics, Super-SBM (slacks-based measure), SDM (spatial Durbin model)

## Abstract

Manufacturing agglomeration (MA) is an important way to achieve high-quality industrial development and promote land use efficiency in China. However, there is a lack of research on the relationship between MA and the green use efficiency of industrial land (GUEIL). Based on the panel data of 279 prefecture-level cities in China, from 2004 to 2019, this study analyzes the spatial and temporal differentiation characteristics of MA and GUEIL, then empirically analyzes the impact of MA on GUEIL. The results show that: (1) during the study period, the national MA levels showed a slight decline, followed by a small increase. In addition, the inter-regional differences are mainly characterized as eastern region > central region > northeast region > western region. (2) The national and regional GUEIL showed a trend of increasing, slightly decreasing, and then increasing again. The overall regional differences in efficiency show the characteristics of: eastern region > central region > western region > northeastern region. (3) At the national scale, MA had a “U-shaped” relationship with GUEIL; at the regional scale, MA had a significant “U-shaped” effect on GUEIL in the northeast, central and western regions, while having a single negative effect in the eastern region. Finally, this study provides suggestions for optimizing the manufacturing structure and improving the GUEIL.

## 1. Introduction

The manufacturing industry is an important factor in measuring a country’s comprehensive national power and plays a pivotal role in the economic development process of various countries. Since China’s reform and opening up, to develop the manufacturing industry, China’s government has been keen to build various types of industrial parks and estates. This has led to the rapid expansion of industrial land. In 2020, China’s industrial land area accounted for 19.4% of the total urban construction land area, far exceeding developed countries, such as the United States (7%), and the international average (10%) [1]. In addition, nearly 5% of China’s urban land resources are idle, and approximately 45% of industrial land is inefficiently used [2]. The development of industrial parks and estates in past decades has created many severe problems, such as idle land, sloppy and inefficient land use, and low output per unit of industrial land [3,4]. In addition, China’s manufacturing industry has been a low-end global manufacturing industry for a long time [5]. High investment, high-energy consumption, and high-emission manufacturing agglomerations have also brought about problems, such as environmental pollution and the excessive consumption of resources [6]. These problems are detrimental to the sustainable use of China’s land resources and high-quality regional development. As a result, optimizing the manufacturing structure and improving GUEIL is a vital way to solve these problems. At present, the green development of the industrial economy has become a global development strategy. The European Commission’s “Industry 5.0: Study in European industry” proposes that “industrial development should pay more attention to social and ecological values”, and should “promote manufacturing green technical innovation.” China has proposed comprehensively improving resource utilization efficiency and vigorously developing the green economy in the “14th Five-Year Plan (2021–2025) for National Economic and Social Development and the Long-Range Objectives through the Year 2035”. In the 20th report of the Chinese Party, General Secretary Jinping Xi mentioned the need “to accelerate the green transformation of the development mode, implement a comprehensive savings strategy, develop a green and low-carbon industry, advocate green consumption, and promote the formation of green low-carbon production methods and lifestyles”. In this context, evaluating GUEIL and analyzing the relationship between MA and GUEIL has essential theoretical and practicable significance for promoting the high-quality development of China’s economy and for constructing an ecological civilization.

At present, there are few direct research results on the impact of MA on GUEIL. Some scholars have conducted a series of studies on the relationship between industrial agglomeration and land use efficiency and between industrial agglomeration and environmental pollution.

In the relationship between industrial agglomeration and land use efficiency, several studies have shown that industrial agglomerations improve land use efficiency [7]. Still, others have also pointed out that the impact of industrial agglomeration on land use efficiency depends on the contrast between the strength of the scale effect and the congestion effect [8]. Moderate industrial agglomeration promotes land use efficiency improvement, while excessive industrial agglomeration inhibits land use efficiency improvement and can even have a negative impact [9]. In the relationship between industrial agglomeration and environmental pollution, three main views have emerged from relevant studies. On the one hand, some scholars believe that industrial agglomeration can effectively alleviate environmental pollution and improve resource utilization efficiency [10,11]. Industrial agglomeration saves production costs and enhances pollution control technology, helping to alleviate environmental pollution through energy-saving and emission-reduction effects. On the other hand, some scholars have suggested that, instead of reducing environmental pollution [12], industrial agglomeration aggravates environmental pollution through congestion effects and concentrated emission effects. Still other scholars believe that the relationship between industrial agglomeration and environmental pollution is non-linear, showing a “U” shape [13], “inverted U” shape [14], and other nonlinear relationships. On the basis of this field of study, scholars have incorporated environmental pollution indicators into the efficiency measurement index system and explored the effects of industrial agglomeration on urban green economic efficiency [15], energy efficiency [16] and industry green efficiency [17]. The above studies have made significant progress; however, industrial land is the spatial carrier of manufacturing industry agglomeration, carrying not only the expected benefits (such as economic output and fiscal revenue), but also non-expected outputs (such as SO_2_, soot, and wastewater). Continuing to focus research solely on the relationship between industrial agglomeration and industrial land use efficiency or environmental pollution will no longer meet the current needs of “comprehensively improving resource use efficiency and vigorously developing the green economy”. With the gradual maturing of the concept of green development, more and more scholars are expressing concern about the green use efficiency of land resources. At present, GUEIL has not been uniformly defined; most previous studies define industrial land use efficiency from the input-output perspective as the degree of the full use of industrial land under a certain output, or as the economic benefits produced under a certain degree of industrial land use. As the concept of the sustainable development of green has gradually become rooted in people’s hearts, scholars have started to consider the ecological costs and ecological benefits of land use and have now incorporated an efficiency evaluation system, which includes factors such as non-expected output. The cost of environmental land production can reveal the economic output capacity [18,19]. Based on these studies, GUEIL can be defined as the economic, social, and ecological benefits generated by industrial land resources under the conditions of considering inputs, such as labor, capital, and resources, and desired outputs, such as industrial value added, and non-desired outputs, such as environmental pollution. Compared with using traditional industrial land use efficiency to maximize economic benefits as the evaluation criteria, GUEIL focuses more on ecological benefits and more accurately reflects the ecological and environmental costs of industrial land use, as well as the level of green development. The core idea of GUEIL can be summarized as obtaining more economic, social, and ecological benefits with less resource loss and environmental pollution. In summary, the following issues remain to be further explored in related studies: first of all, existing studies have mostly examined manufacturing agglomeration development from a static perspective. However, manufacturing agglomeration is not simply manifested by the small-to-large area of industrial land in cities and the increase in the number of enterprises but also by dynamic changes, such as the elimination of superior and inferior enterprises and industrial transformation and upgrading. Secondly, the existing research on the relationship between industrial agglomeration and land use efficiency does not give enough consideration to the ecological benefits of land use. Current research on the relationship between industrial agglomeration and green land use efficiency is more in line with the current value orientation of sustainable land use in China. Finally, the current research on industrial agglomeration and environmental pollution has mostly considered the perspectives of cities and industries but lacks analysis from the perspective of land use. As such, these studies have ignored the important role of industrial land as a carrier of manufacturing agglomeration and scarce input resources in manufacturing agglomeration and industrial green development. In addition, the research on MA and GUEIL can play a complementary role to the above research. Based on the above deficiencies, this study analyzes the spatial and temporal variation characteristics of MA and GUEIL across China and each region, using data from 279 prefecture-level and above cities in China, from 2004–2019. This research empirically investigates the effect of MA on GUEIL, using the SDM model to provide a scientific basis for promoting the transformation and upgrading of MA and improving GUEIL in China.

## 2. Materials and Methods

### 2.1. Empirical Methodology

In reality, GUEIL has obvious spatial correlations in various places, because of the inter-regional factor flow, resource allocation, and industrial transfer. Therefore, this study plans to examine the effect of MA on GUEIL by using a spatial econometric model.

Spatial econometric models include three forms of spatial Durbin model (SDM), spatial error model (SEM), and spatial lag model (SLM). This study first uses Moran’s index to test the spatial correlation of GUEIL. Next, the Lagrange multiplier (LM) test, likelihood ratio (LR) test, Wald test, Hausman test, and joint significance test were used to determine the specific form of the spatial econometric model. The test results indicate that the time-space two-way fixed spatial Durbin model is the optimal model for this study. The equation of the model is as follows:(1)$$\begin{array}{c} \ln eil_{it}=\alpha_{i}+\delta W_{n}\ln eil_{it}+\beta_{1}agg_{it}+\beta_{2}agg_{it}{}^{2}+\beta_{3}X_{it}+\\ \theta_{1}W_{n}agg_{it}+\theta_{2}W_{n}agg_{it}{}^{2}+\theta_{3}W_{n}X_{it}+u_{i}+v_{t}+\varepsilon_{it} \end{array}$$

In the above equation, $\ln eil_{it}$ denotes GUEIL, and $agg_{it}$ denotes the level of MA. The squared term of the MA value is also added to the model to examine the quadratic relationship between agglomeration and efficiency. Next, $W_{n}$ is the spatial weight matrix, δ is the spatial autoregressive coefficient, β is the regression coefficient of the explanatory variable, θ is the regression coefficient of the spatial lagged term of the explanatory variable, $u_{i}$ is the region fixed effect, $v_{t}$ is the time fixed effect, $\varepsilon_{it}$ is the random error term, $\alpha_{i}$ is the constant term, and *i* and *t* denote city and year, respectively.

### 2.2. Variable Setting

(1)Explained variable: green use efficiency of industrial land

The Super-SBM model, first proposed by Tone K [20], can solve the input-output shortage problem and effectively address the shortcomings of previous SBM models that cannot continue to evaluate and rank the effective units. Therefore, this study measures GUEIL through the Super-SBM model. The measures of GUEIL contain input indicators, desired output indicators, and non-desired output indicators. The selected input and output indicators are shown in Table 1.

(2)Core explanatory variable: manufacturing agglomeration

In this paper, location entropy is used to measure the MA level, which can eliminate the scale difference between regions and objectively reflect the spatial distribution of industrial agglomeration [15]. The calculation formula of the MA degree is as follows: (2)$$agg_{it}=\frac{e_{it}/\sum_{i}e_{it}}{\sum_{s}e_{it}/\sum_{i}\sum_{s}e_{it}}$$

In the above formula, *i* denotes city, *t* denotes time, $e_{it}$ denotes the number of people employed in manufacturing in city *i*, $\sum_{i}e_{it}$ denotes the total number of people employed in manufacturing, $\sum_{s}e_{it}$ denotes the number of people employed in urban units in city *i*, and $\sum_{i}\sum_{s}e_{it}$ denotes the number of people employed in urban units nationwide.

(3)Control variables

Considering the relevant literature, the control variables selected for this study include environmental regulation [21], economic development level, industrialization degree, opening degree, technology input level, government intervention level, informationalized level, eco-environmental endowment, and regional innovation level. Table 2 presents the results of the descriptive statistics for the variables.

### 2.3. Data Sources

The data used in this study include: (1) employee numbers in urban manufacturing, the industrial emissions of three wastes, science and technology expenditures, public finance expenditures, international Internet users, and greening coverage of built-up areas. These data are obtained from the China City Statistical Yearbook (2004–2020). (2) Urban industrial land area data are from China Urban Construction Statistical Yearbook (2004–2020). (3) Number of city invention patents that have been granted are taken from the China Research Data Service Platform (CNRDS). (4) The number of urban unit employees, year-end population, total assets of industrial enterprises above the scale, industrial value added, GDP, GDP per capita, total import and export trade, GDP index, GDP per capita index, and consumer price index data are all obtained from provincial and municipal statistical yearbooks (2004–2020). (5) The data regarding the number of employees in the national manufacturing industry and the number of employees in urban units nationwide are from the China Statistical Yearbook (2004–2020). The total assets of industrial enterprises above the scale and industrial value added are converted into comparable prices through the resident consumption index, based on 2003 measurements. The GDP and GDP per capita figures are adjusted to comparable values by using the GDP index and GDP per capita index, based on 2003 values. Total import and export trade is converted at the average exchange rate between the U.S. dollar and the RMB for the calendar year. Most missing data values were supplemented by the China Regional Economic Statistical Yearbook (2004–2014), the China Marine Statistical Yearbook, provincial and municipal statistical yearbooks, economic yearbooks, and the National Economic and Social Development Statistical Bulletin. This study also uses linear interpolation methods for individual missing data that could not be obtained from the yearbooks.

The research object of this paper is Chinese cities at the prefecture level and above, from 2004–2019. Due to the lack of data in provinces and cities such as Tibet Autonomous Region, Luliang, Yingkou, and Zhongwei, as well as the adjustment of administrative divisions in cities such as Haidong, Turpan, and Danzhou during the study period, 279 cities were finally selected as the research object. For further discussion, the country was also divided into northeastern, eastern, central and western regions, according to the differences in geographical areas and economic development.

## 3. Results

### 3.1. Spatial and Temporal Variation Characteristics of MA and GUEIL

#### 3.1.1. Measurement Results and Spatio-Temporal Analysis of MA

Figure 1 shows the national and regional MA measurement results, and Figure 2 reflects the spatial distribution of MA in Chinese cities. According to Figure 1 and Figure 2, the average value of the national MA index was 0.89 during the study period, with an overall trend of a slight decrease, followed by a slight increase. The average values of the MA index in the eastern, central, northeastern, and western regions were 1.21, 0.86, 0.69, and 0.67, respectively, showing the characteristics of eastern region > central region > northeastern region > western region, which is roughly equivalent to the gradient difference of economic development. The MA level in the eastern region is substantially ahead of other regions. Significantly, the MA level of Huizhou, Shenzhen, Quanzhou, and other cities in the Pearl River Delta region and Suzhou and Wuxi in the Yangtze River Delta region has always remained at a higher level. The reason for this finding may be that the eastern region, as a priority, enjoys the reform and opening up policy dividend and has the first-mover advantage, which gives priority to the manufacturing industries transferred from developed regions and countries. Thus, the manufacturing industries in the region have a tremendous economic leadership agglomeration advantage [22]. The MA level in the central region showed a significant upward trend year-by-year, and Jingmen, Chuzhou, Ji’an and other prefecture-level cities and their surrounding cities have the most obvious degree of MA. This finding may be closely related to a series of policies implemented by the central government to promote the rise of the central region, as well as the policy of promoting the gradual transfer of manufacturing industries from the eastern region to the central and western regions [23]. The MA level in the west and northeast has been declining year-by-year and was lower than the national average during the study period. These areas include Harbin, Qiqihar, Jilin and other cities in the northeast and Lanzhou, Zhangye, Leshan and other cities in the west. The reasons for this finding may be that: (1) the western region lacks location advantages; the overall level of economic development is not high, and the area is not very attractive to industrial investment. (2) Although the northeast region has certain location advantages and a good industrial base, the region has encountered some setbacks in economic development and difficulties in industrial transformation in recent years. The result has been a large number of young and strong people flowing to the central and eastern regions. The above factors have restricted the development of industrial clustering in China’s western and northeastern regions.

#### 3.1.2. Measurement Results and Spatio-Temporal Analysis of GUEIL

Figure 3 reflects the changes in GUEIL in China and its regions from 2004 to 2019, while Figure 4 shows the spatial distribution of GUEIL in Chinese cities during those years. One can find that the overall level of GUEIL nationwide shows a trend of first increasing, then slightly decreasing and then increasing. There was a general trend of increasing efficiency from 2004–2012 and a slight decrease in efficiency from 2012–2015. The reason may lie in the following: the large-scale construction of industrial parks and industrial zones in various regions has resulted in the rapid expansion of industrial land, but the lack of industrial imports has left much industrial land idle and inefficient. In addition, the low price of land resources acquired by enterprises will inevitably lead to the substitution of land resources for other production factors, such as technology and capital. This will reduce the level of technical efficiency and technological progress, thus reducing the GUEIL. With the “2015 Industrial Green Development Special Action Implementation Plan” and the “Industrial Green Development Plan (2016–2020)”, launched in 2016, China’s Ministry of Industry and Information Technology provided a guide to direct the green development of industry around the country. Therefore, the GUEIL, both nationwide and in all regions, significantly improved in the years 2015–2019.

In recent years, the GUEIL has varied significantly among regions in the country, showing the characteristics of eastern region > central region > western region > northeastern region. This is basically consistent with the findings of a number of scholars [24]. In most of the studied years, the GUEIL in the eastern region was above the national average. Jinhua and Ningde, with GUEIL values higher than 0.8 in 2004, are in the eastern coastal region. The GUEIL in most cities in the Pearl River Delta, Yangtze River Delta city cluster, and Fujian Province has increased significantly in the past 16 years. This finding indicates that cities in the eastern region have resources, talents, and policy leadership in terms of promoting green industrial development. The average GUEIL value in the central area is basically consistent with the national average. Most cities in Shanxi, Anhui, Henan, Hubei, and Hunan have improved their GUEIL. Among them, Kaifeng, Changde, Changsha, and other cities have significantly enhanced their GUEIL, indicating that the central region has the role of regional coordination in geographic and economic development. In addition, the central region’s cities’ GUEIL is improving yearly, so this will be the main region for advanced and green industry development in the future. The GUEIL in the western region has lagged behind the national average most of the time; this has been caused by backward industries, technology, and a talent shortage. However, in recent years, the western region has taken in many industries and technology transfers from the east and central regions, optimizing local industries’ production technology and management level. As a result, the GUEIL has grown faster. The northeast region had the lowest industrial land use efficiency during the study period, because it is a traditional heavy industrial base with a single industrial structure that relies on coal, steel, and other industries.

#### 3.1.3. Comprehensive Analysis of MA and GUEIL

Four dimensions can be generated by combining MA level and GUEIL. An MA level lower than the average MA value is “low agg”, while an MA value greater than the average MA value is “high agg”. A GUEIL level that is lower than the average GUEIL value is “low eil”, while a GUEIL value greater than the average GUEIL value is “high eil”. Cities in Quadrant I have high MA levels and high GUEIL. Cities in Quadrant II have low MA levels and high GUEIL. Cities in Quadrant III have low MA levels and low GUEIL. Cities in Quadrant IV have high MA levels and low GUEIL (see Figure 5 below).

The values of the above two variables in 2004, 2009, 2014, and 2019 are presented in Figure 6. From Figure 6, one can find that, in 2004, the number of cities in Quadrants I–IV were 11, 32, 122, and 114, respectively. Cities in Quadrants III and IV dominated. The cities in Quadrant III are mainly located in the northeast and central regions. The cities in Quadrant IV are mainly located in the northeast and eastern regions. The cities in Quadrant II are mainly located in the western and central regions, and the cities in Quadrant I are mainly located in the eastern coastal regions. From 2004 to 2019, the city types in the northeast region mainly changed from Quadrant III to Quadrant II. The city types in the east region mainly changed from Quadrant IV to Quadrant I. The city types in the central region changed from Quadrant III and Quadrant IV to Quadrant II and Quadrant I, and the city types in the west region mainly changed from Quadrant III and Quadrant IV to Quadrant II. In 2019, the number of cities in Quadrants I–IV were 80, 134, 29, and 36, respectively. Cities in Quadrant II and Quadrant I dominated. The cities in Quadrant I are mainly distributed in the eastern and central regions. The cities in Quadrant II are mainly distributed in the central and western regions. The cities in Quadrant IV are mainly distributed in the northeast and central regions, and the cities in Quadrant III are mainly distributed in the northeast and western regions. The comprehensive analysis shows that, during the studied years, the sample cities generally showed a trend of increasing manufacturing concentration and improving their GUEIL. This finding indicates that Chinese cities are generally in the process of green transformation and upgrading manufacturing industries; partial results have already been achieved. 

A further analysis of the “agg-eil” typology of Chinese cities from 2004–2019 shows that some cities have distinct typological characteristics. Typical cities in Quadrant I include Shenzhen, Xiamen, Ningbo, and Xiangyang. These cities’ manufacturing industries are still dominant, but there has been a successful transformation from labor-intensive manufacturing to new and high-end manufacturing industries. Examples include knowledge-intensive and technology-intensive industries, so these industries have high GUEIL. Typical cities in Quadrant II include Beijing, Chengdu, Haikou, Jinhua, and Yulin. The service industry in this type of city is developing faster, and manufacturing is gradually becoming a disadvantageous industry. In addition, this type of city generally attaches importance to environmental quality and is based on developing a high-end and green manufacturing industry. Therefore, the city’s industrial land is more efficient for green use. There are two types of cities in Quadrant III. One type is comprised of cities with manufacturing industries that do not dominate the industrial system, such as Zhangjiakou, Fuxin, and Guilin. The other type is made up of cities with labor-intensive, low-end manufacturing industries that rely on resource advantages or geographical advantages. Transforming and upgrading these industries is more challenging, causing urban manufacturing industries to remain at the low-end of the value chain and restricting GUEIL. Jiayuguan, Huanggang, Anshan and Jilin represent cities in Quadrant IV. This type of city is mainly specified as a manufacturing city; these are the places in which labor-intensive and capital-intensive manufacturing industries are located. The congestion effect caused by a large number of low-end manufacturing industries being concentrated in the same area inhibits the improvement of GUEIL. 

### 3.2. The Effect of MA on GUEIL

#### 3.2.1. Model Testing and Identification

The first definition of geography holds that everything has an interaction, and similar things are closer [25]. Anti-distance weight is the most in line with geography’s first-defined space weight matrix. The larger is the distance between the space unit, the smaller is the weight. This study uses the countdown of the geographical distance between the city to build an anti-distance weight matrix. The authenticity of the inspection space of Moran’s index is used to review Table 3. The significance test was passed in all years of the study period, except for 2011. The green use efficiency of urban industrial land in China shows a positive spatial correlation, and the spatial correlation generally decreased first and then increased. The results of the tests in Table 4 indicate that the time-space dual fixed space Durbin model is optimal for the empirical analysis.

#### 3.2.2. Analysis of National-Scale Regression Results

According to the estimation results of Model (1) in Table 5, one can find the first and secondary coefficients of the manufacturing gathering are −0.4963 and 0.1046 all of which are significant at the 1% level. This finding indicates the impact of MA on GUEIL in Chinese cities is “U-shaped” at the national level. The initial MA generally appears at a low level of economic development or a lower level of economic development and is manifested as the improvement of the level of MA to reduce GUEIL. The main reason is that, under the GDP-oriented official assessment mechanism, local governments are prone to “bottom-up competition”; they tend to attract investment to achieve rapid economic development by using low-priced industrial land concessions and relaxed environmental controls [26]. At this stage, the manufacturing industry is “artificially manufactured,” leading to serious enterprise homogeneity and a lack of “vertical” and “horizontal” industrialized division in the labor system and industrial connection. The enterprises in the agglomeration area adopt the production mode of land (instead of capital or technology), and they have a low proportion of investment funds for scientific research and talent. Therefore, in the studied years, the decline of green innovation ability and the excessive consumption of industrial land resources in the agglomeration resulted in inefficient industrial land use and serious environmental pollution problems, thus reducing the GUEIL. With the continuous development of the economy, the manufacturing industry continues to develop, and the degree of agglomeration continues to escalate. The agglomeration of the manufacturing industry has turned to promoting the green use of industrial land to improve efficiency. The main reason is that the environmental pollution problem arising from the concentration of manufacturing industries in urban areas is endangering people’s health and, in turn, sustainable economic and social development. Under the guidance of the ecological civilization construction system, the central government has continuously improved the mechanism used to assess officials, shifting from GDP development to green development orientation. China’s local governments have continually strengthened environmental regulation, which is conducive to reducing the non-expected outputs carried by industrial land. Moreover, in recent years, China has implemented a series of policies, such as the national minimum price standard for industrial land transfer, land conservation and use policies, guidelines for the implementation of industrial land use policies and the establishment of a green development economic system. These policies have effectively curbed the excessive expansion of industrial land and improved the intensive use of industrial land. Furthermore, the enterprises in the agglomerations are continuously being eliminated, transformed, and upgraded in the competition. The connection between upstream and downstream industries has been strengthened. Agglomerations with high-tech manufacturing enterprises, represented by Zhongguancun Science Park, Shenzhen High-Tech Industrial Park and Suzhou Industrial Park, have emerged, representing the development of MA in various places moving towards the development of high-level GUEIL. Furthermore, compare the “inflection point” of manufacturing aggregates and the gathering level of cities in multiple cities. By 2019, the MA level of most cities is on the left side of the “inflection point”; this is still at the stage of curbing the GUEIL.

Among the control variables, the smaller is the value of environmental regulation, the greater is the intensity of environmental regulation, which has a significant promotion effect on the improvement of GUEIL. China’s central and local governments can effectively prompt industries to improve their production technologies, increase the tail-end treatment of pollutants, and internalize external costs by raising the enterprises’ environmental awareness, thus enhancing the efficiency of the green use of regional industrial land. In addition, GDP makes a significant positive contribution to GUEIL. When the level of economic development is higher, the foundation of industrial development and the market mechanism are usually better. Additionally, the higher are people’s demands for ecological and environmental quality, the more pressure the government will face in terms of environmental management, the more investment will be made in green science and technology research, and the more intense will be the competition among enterprises. All of this will serve to eliminate backward production capacity and encourage the green transformation and development of enterprises. The level of industrialization has a significant positive effect on GUEIL. Cities with a high degree of industrialization also have a correspondingly high level of industrial-intensive production, which is conducive to the inter-industry and intra-industry division of labor and cooperation, thus improving industrial land use efficiency. The coefficient of the influence of the degree of external openness on GUEIL is significantly positive. Cities with a high degree of external openness have more opportunities to establish contacts with advanced production technology and with those with green management experience at home and abroad. This makes it easier to form technology and knowledge spillover effects and is conducive to improving GUEIL. The coefficients of the impact of the level of science and technology inputs and the level of regional innovation are significantly positive. An increase in R&D investment can effectively enhance enterprises’ research and innovation capability, thus accelerating the innovation of industrial production technology and pollution control technology. This will effectively reduce production costs and energy consumption, reduce undesired output, and contribute to the improvement of GUEIL. The degree of government intervention has a significantly negative effect on GUEIL. Government intervention will affect the flow of labor and capital, which is not conducive to the division of labor and technological innovation among enterprises. The government’s pursuit of land finance will produce a “race to the bottom” effect, leading to the mismatch of land resources and the inefficient use of industrial land, ultimately inhibiting GUEIL. The level of informatization makes a significant positive contribution to GUEIL. The informationalized level can effectively improve the level of resource, energy, and environment management, accelerate the digital transformation of industrial production methods, and enhance GUEIL.

#### 3.2.3. Regional Heterogeneity Analysis

In the process of MA and industrial development, the effect of MA on GUEIL will show regional heterogeneity, being influenced by the different geographical locations of cities, economic development levels, and policy responses.

The regression results in Table 5 show a significant “U-shaped” effect of MA on GUEIL in northeastern, central, and western regions of China, consistent with the national scale estimation results. In contrast, the “U-shaped” effect in the eastern region is not obvious. Here, MA has a single negative effect, probably because the eastern region of China has a high level of specialized concentration. Still, industrial convergence is more obvious, which is not conducive to industrial structure upgrading and transformation [27]. Moreover, with the gradual transfer of low-value-added and high-energy-consuming manufacturing enterprises from the eastern region to other regions, problems such as the conversion of new and old dynamics are prominent [23]. These problems make promoting GUEIL very challenging. According to the results of the comparison between the average value of MA and the inflection point value of cities in each region in 2019, one can find that the degree of MA in most cities in these regions had not reached the inflection point value and was still at the stage of the negative impact of MA on the green use efficiency of urban industrial land.

#### 3.2.4. Robustness Tests

The study period was divided into three periods (2004–2009, 2010–2014 and 2015–2019) for robustness testing (see Table 6). The regression results show a “U-shaped” effect of MA on GUEIL in different periods. This is consistent with the previous estimation and indicates that the empirical results are robust.

In addition, to overcoming the possible endogeneity problem of the model, the study also uses SYS-GMM to estimate the model and validates it by comparing it with a fixed-effects model. The estimated results of SYS-GMM and the fixed-effects model are consistent with the previous estimates, and the empirical results are robust (see Table 7 below).

## 4. Discussion

This paper firstly explores the spatial and temporal variation characteristics of MA and GUEIL in China. The manufacturing industry in the eastern region is found to have played a greater economic leadership role in agglomeration; the MA level in the central region showed a significant upward trend year-by-year, and the MA level in the western and northeastern regions decreased year by year. These findings reflect the current trend of transferring the manufacturing industry from the eastern region to the central and other regions in China. Meanwhile, the western and northeastern regions have had difficulties in industrial transformation and are burdened with weak industrial attractiveness. This has resulted in the backward development of the manufacturing industry in these regions. The above findings are similar to the findings of Fu et al. [28]. Some scholars have found that obvious regional differences exist in GUEIL, with the average in the eastern region being higher than that in the central and western regions. From a spatial perspective, the GUEIL has experienced a trend of first decreasing and then continuously increasing [24]. This finding is similar to the spatial and temporal divergence results of the GUEIL in this paper, indicating that, under China’s green development strategy, the country’s industrial land is changing from rough utilization to green and sustainable utilization. Second, this paper shows that the overall MA in China has a “U-shaped” relationship with GUEIL. There is also a significant “U-shaped” effect of MA on GUEIL in the northeast, central and western regions, while MA in the eastern region has a single negative effect. The effect of MA on GUEIL varies in line with the development stage of manufacturing agglomeration, and there is regional heterogeneity. Several scholars have also conducted related studies and obtained different conclusions. Han et al. found that, among the secondary industries, manufacturing agglomeration in China has a positive effect on land use efficiency [7]. Zeng argued that an increase in the secondary industry suppresses land green use efficiency [29]. The former land use efficiency finding only considers the economic output efficiency of land use; the latter considers the undesired output of land use but based on cross-sectional data analysis. The conclusions obtained from both have limitations. In addition, some scholars have explored the impact of MA on green economic efficiency. Although such studies are not based on the perspective of industrial land, they have a certain reference value. Yuan et al. found that MA could promote green development efficiency in the upper reaches of the Yangtze River but hinder the improvement of green development efficiency in the middle and lower reaches [30]. Ji et al. found that both labor-intensive manufacturing agglomeration and capital-intensive manufacturing agglomeration will hinder the improvement of the local green economic efficiency level. Meanwhile, high-tech manufacturing agglomeration is conducive to the improvement of the green economic efficiency level [31]. These findings indicate that heterogeneity exists in the impact of MA on green economic efficiency in different regions and different agglomeration levels, and this view is also applicable to this paper.

This research makes several contributions. First, this study expands the concept of GUEIL. Compared with traditional industrial land use efficiency, which is evaluated by maximizing economic benefits, the interpretation of the concept of GUEIL in this paper focuses more on ecological benefits. This approach more accurately reflects the ecological and environmental costs of industrial land use and the level of green development. Second, this study provides a new perspective for understanding the determinants of GUEIL. Industrial land is the carrier of MA and a scarce input resource, MA is closely related to GUEIL. Previous studies have mainly focused on the influence of government and market behaviors on industrial land use efficiency. In contrast, this study examines the effect of MA on GUEIL, which is conducive to enriching existing research related to the influencing factors of GUEIL. Third, this paper analyzes the spatial analysis characteristics of MA and GUEIL and the influence effects between them from two perspectives: nationwide and regional in China. The objective is to provide a scientific basis for promoting the transformation and upgrading of MA and improving GUEIL in different regions of China. The results also serve as a reference for countries and regions with similar conditions to China.

The shortcomings of this study are as follows: firstly, the measurement system for GUEIL is not perfect. The system is subject to data accessibility and fails to include green indicators, such as low energy consumption and low carbon. Secondly, this study fails to further subdivide the manufacturing industry and lacks research from the perspective of specialized and diversified MA. Future research can improve the system used to measure GUEIL and can also analyze the effect of specialized and diversified MA, especially the effect of low-end and high-end MA patterns on GUEIL.

## 5. Conclusions

This study uses the panel data of 279 prefecture-level and above cities across China, from 2004 to 2019, to analyze the spatial and temporal variation characteristics of MA and GUEIL across China, both nationally and regionally. The aim is to empirically investigate the effect of MA on GUEIL. The results show that (1) the national MA levels showed a slight decline during the studied period, followed by a small increase; the inter-regional differences are mainly characterized as eastern region > central region > northeast region > western region. (2) The national and regional GUEIL showed a trend of increasing, slightly decreasing, and then increasing. The overall regional differences in efficiency show the characteristics of eastern region > central region > western region > northeastern region. (3) At the national scale, MA has a “U-shaped” relationship with GUEIL. At the regional scale, MA has a significant “U-shaped” effect on GUEIL in the northeast, central and western regions, while in the eastern region, MA has a single negative effect.

This study has a number of policy implications. (1) Promote the development of MA “according to local conditions” in each region. The eastern region should optimize the land layout and structure of the agglomeration on the limited industrial land. Full use should be made of the advantages in terms of capital, technology, and human energy to empower green manufacturing and to promote the efficient use of industrial land. Central and western regions should accumulate “building nests and phoenixes” in the regions’ resources and labor, restrict high pollution manufacturing industries, and actively accept manufacturing enterprises that meet regional development standards in order to create new high-quality MA. The northeast region should rely on the strategy of “revitalizing the old industrial bases in the region” to explore new paths to world-class manufacturing clusters. This could be achieved by transforming and upgrading traditional manufacturing industries to advanced equipment manufacturing, petrochemical and fine chemicals, metallurgical new materials, and other high-end manufacturing industries. (2) Establish and improve a “green” industrial economy evaluation system, guiding local governments to change the traditional concept of being “GDP-oriented”. Consider including industrial land energy use, resource use, ecological factors, environmental protection, and other aspects of green indicators into the industrial land saving and intensive use assessment system. Additionally, include green economic development indicators, such as GUEIL, into the government performance assessment mechanism. These steps should be taken to reverse the vicious competition situation of local economic development at the expense of the environment. (3) Enhance the awareness of green development of manufacturing enterprises and actively carry out green production technology changes. First, strengthen environmental regulation and supervision and strictly supervise and control the emission behavior of enterprises. Second, use fiscal means, such as imposing higher environmental protection taxes on polluting enterprises and providing financial subsidies for green development enterprises. Third, combine regional resource endowments and related plans; formulate manufacturing investment access and forced exit lists. Fourth, promote technical cooperation among colleges and universities-research institutions-enterprises. The aim should be to encourage enterprises to invest in developing green production technology and pollutant emission reduction technology.

## Figures and Tables

**Figure 1 ijerph-20-01575-f001:**
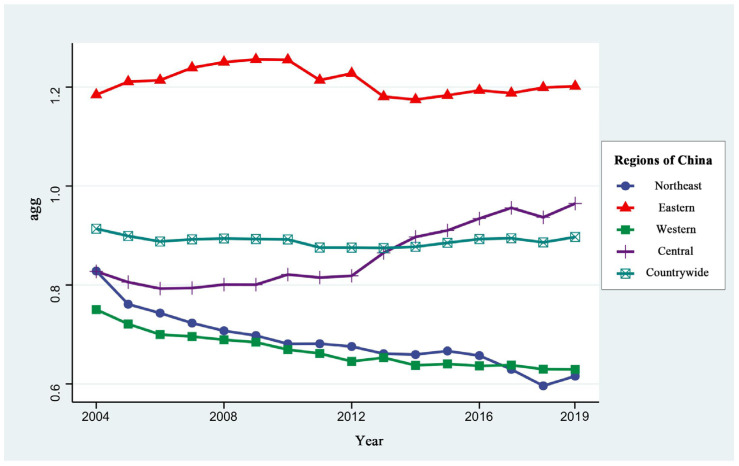
Changes of MA in China and its regions from 2004–2019.

**Figure 2 ijerph-20-01575-f002:**
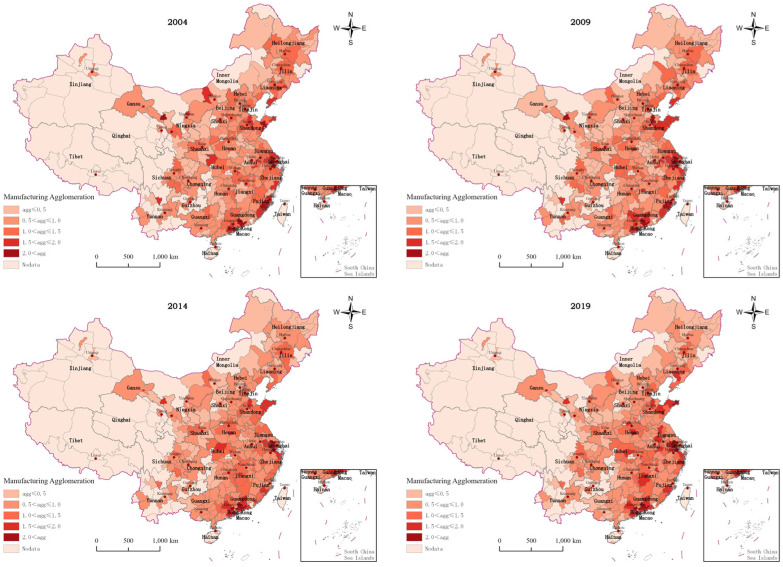
Spatial distribution of MA in Chinese Cities from 2004–2019.

**Figure 3 ijerph-20-01575-f003:**
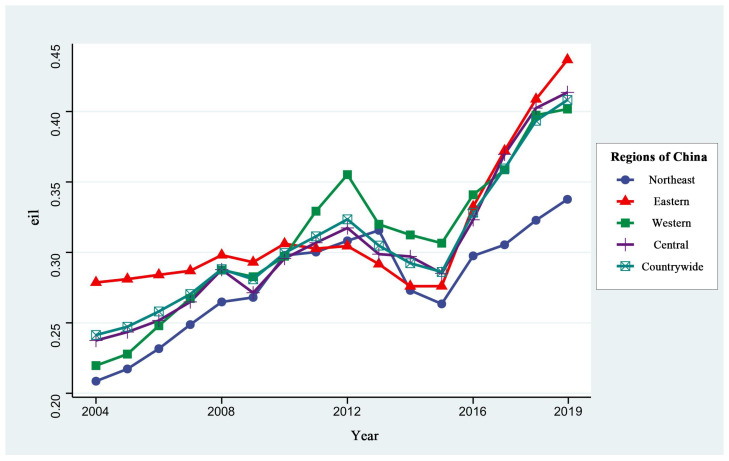
Changes in GUEIL nationwide and by region from 2004–2019.

**Figure 4 ijerph-20-01575-f004:**
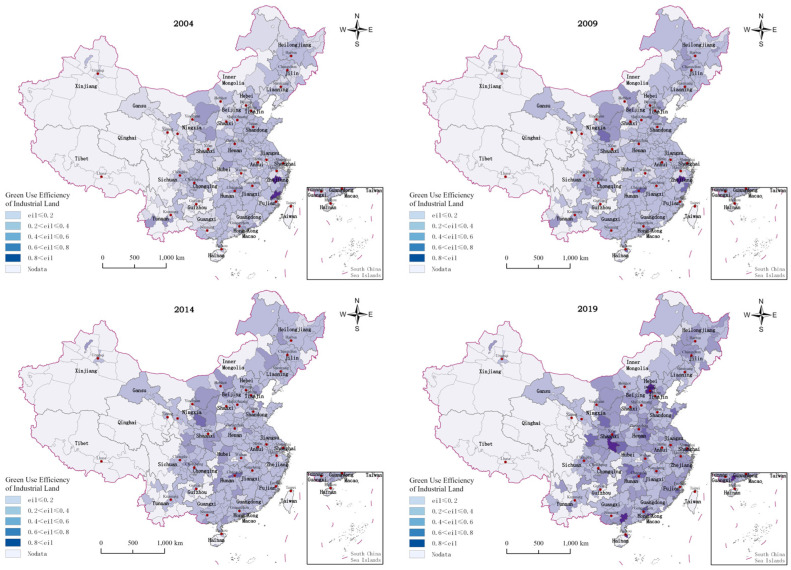
Spatial distribution of GUEIL in China from 2004–2019.

**Figure 5 ijerph-20-01575-f005:**
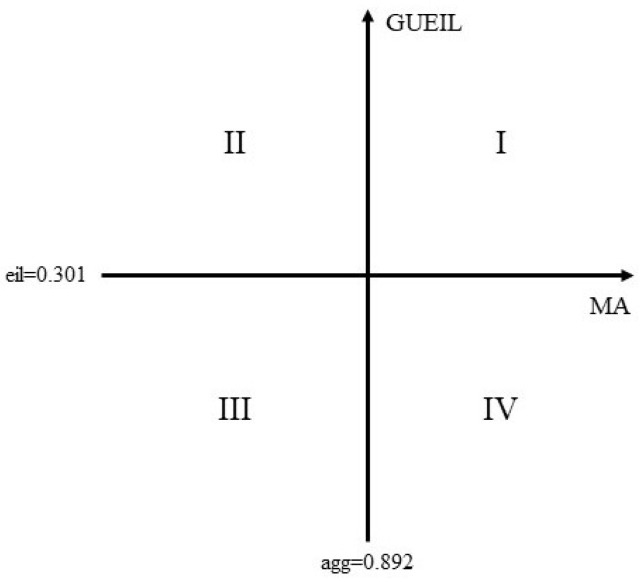
Comprehensive Analysis of MA and GUEIL.

**Figure 6 ijerph-20-01575-f006:**
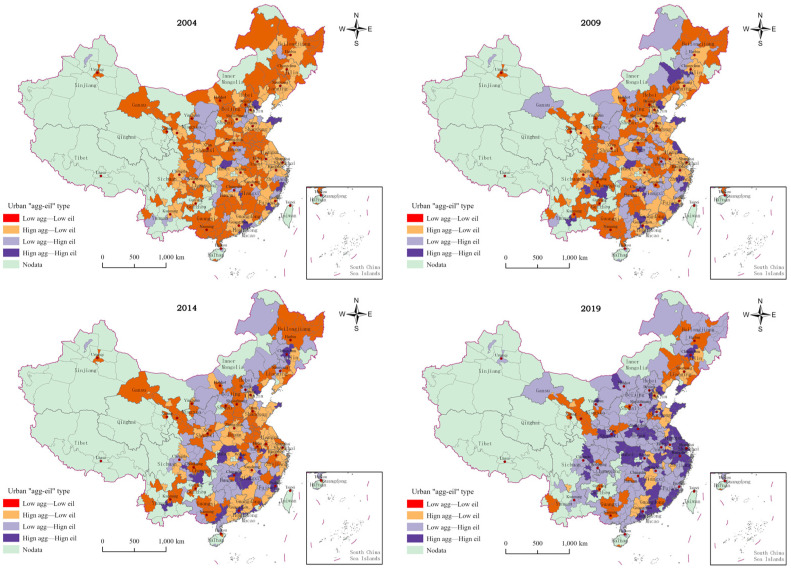
Classification of “agg-eil” types of Chinese cities from 2004–2019.

**Table 1 ijerph-20-01575-t001:** Input-output indicators for measuring GUEIL.

Indicator Type	Element Name	Indicator Name
Input Indicators	Capital Investment	Total assets of industrial enterprises above average per land(Million Yuan/km^2^)
	Labor input	Number of manufacturing workers per land (million people/km^2^)
	Land input	Industrial land area (km^2^)
Output Indicators	Expected output	industrial value added per land (billion yuan/km^2^)
	Non-desired outputs	industrial sulfur dioxide emissions per land (tons/km^2^)
		industrial wastewater discharge per land (million tons/km^2^)
		industrial smoke (dust) emissions per land (tons/km^2^)

**Table 2 ijerph-20-01575-t002:** Descriptive statistics results.

Variables	Variable Name	Measurements	Mean	Min	Max	Sample Number
lneil	green use efficiency of industrial land	Super-SBM model	−1.252	−3.135	0.286	4464
agg	manufacturing agglomeration	location entropy	0.889	0.022	3.189	4464
lnep	environmental regulation	index of “industrial waste”	−3.062	−14.526	3.558	4464
lnpgdp	economic development level	GDP per capita	10.076	7.603	13.245	4464
is	Industrialization degree	share of secondary industry in GDP	0.473	0.107	0.910	4464
od	opening degree	Total import and export trade/ regional GDP	0.204	0.000	8.134	4464
si	technology input level	science and technology expenditure/public finance expenditure	0.013	0.000	0.131	4464
gil	government intervention level	public finance expenditure/GDP	0.171	0.040	1.027	4464
ifi	Informationalized level	number of international Internet users/year-end population	0.160	0.001	1.987	4464
gc	eco-environmental endowment	greening coverage of built-up areas	0.376	0.004	0.696	4464
ri	regional Innovation Level	number of invention patents granted/total population at the end of the year	0.914	0.000	48.132	4464

**Table 3 ijerph-20-01575-t003:** Moran’s I change of GUEIL from 2004 to 2019.

**Year**	**2004**	**2005**	**2006**	**2007**	**2008**	**2009**	**2010**	**2011**
Moran’s I	0.078 ***	0.078 ***	0.062 ***	0.044 **	0.051 ***	0.043 **	0.040 **	0.023
sd statistics	4.403	4.367	3.509	2.542	2.930	2.502	2.353	1.448
**Year**	**2012**	**2013**	**2014**	**2015**	**2016**	**2017**	**2018**	**2019**
Moran’s I	0.033 *	0.028 *	0.067 ***	0.060 ***	0.057 ***	0.066 ***	0.092 ***	0.107 ***
sd statistics	1.953	1.725	3.774	3.429	3.296	3.766	5.160	5.921

Note: * represents significance at the 10% level, ** represents significance at the 5% level, *** represents significance at the 1% level.

**Table 4 ijerph-20-01575-t004:** Test results of model settings form.

Test Method	Test Statistic Results	*p* Value	Test Conclusion
LM-Lag	81.106	0.0000	Can choose SEM model
Robust LM-Lag	3.858	0.0490	
LM-Error	1047.219	0.0000	Can choose SAR model
Robust LM-Erro	969.971	0.0000	
LR-SDM-SEM	244.98	0.0000	Compared with SEM and SAR models, choosing SDM models is better
LR-SDM-SAR	201.23	0.0000	
Wald-SAR	73.14	0.0000	
Wald-SEM	70.98	0.0000	
Hausman	−208.80	-	Choosing the SDM model, the fixed effect model is better
Time Effect	4392.19	0.0000	Choosing the SDM model, the dual fixed model of time and space is better
Spatial effects	160.87	0.0000	

**Table 5 ijerph-20-01575-t005:** Regression results of the effect of MA on GUEIL across China and different regions.

Variables	eil				
	Countrywide	Eastern	Central	Western	Northeast
	(1)	(2)	(3)	(4)	(5)
agg	−0.4963 ***	−0.2461 **	−0.4176 ***	−0.9803 ***	−0.7956 ***
	(0.0384)	(0.0765)	(0.0830)	(0.0781)	(0.1411)
agg2	0.1046 ***	0.0113	0.0756 *	0.4039 ***	0.3242 ***
	(0.0145)	(0.0239)	(0.0393)	(0.0388)	(0.0792)
lnep	−0.0418 ***	−0.0640 ***	−0.0398 ***	−0.0267 ***	−0.0525 ***
	(0.0030)	(0.0061)	(0.0054)	(0.0055)	(0.0080)
lnpgdp	0.1693 ***	−0.1566	0.0714	0.3369 ***	0.3869 **
	(0.0405)	(0.1099)	(0.0751)	(0.0630)	(0.1192)
is	1.1312 ***	0.8055 ***	0.8245 ***	1.3369 ***	0.6410 ***
	(0.0678)	(0.1868)	(0.1283)	(0.1058)	(0.1836)
od	0.0534 ***	0.0678 ***	−0.1644 *	−0.0062	0.0893 *
	(0.0142)	(0.0153)	(0.0920)	(0.0532)	(0.0529)
si	1.9923 ***	2.2173 **	−1.5584 **	3.5111 **	2.5782
	(0.4505)	(0.7086)	(0.7104)	(1.5000)	(1.7332)
gil	−0.6275 ***	−0.9827 ***	−1.0132 ***	−0.4445 ***	−1.7832 ***
	(0.0816)	(0.2709)	(0.2431)	(0.1111)	(0.1904)
ifi	0.0687 **	0.0205	0.6316 ***	0.0590	0.2553 **
	(0.0335)	(0.0427)	(0.1249)	(0.0713)	(0.1211)
gc	−0.0014	−0.1583	−0.1901 **	0.0003	0.3994 **
	(0.0495)	(0.1009)	(0.0829)	(0.0861)	(0.1499)
ri	0.0184 ***	0.0164 ***	0.0071	0.0633 ***	−0.0303 *
	(0.0019)	(0.0022)	(0.0081)	(0.0119)	(0.0180)
City Fixed	YES	YES	YES	YES	YES
Time fixed	YES	YES	YES	YES	YES
N	4464	1360	1264	1312	528

Note: Values in parentheses are standard errors, * represents significance at the 10% level, ** represents significance at the 5% level, and *** represents significance at the 1% level.

**Table 6 ijerph-20-01575-t006:** Robustness test results.

Variables	2004–2009	2010–2014	2015–2019
	(6)	(7)	(8)
	lneil	lneil	lneil
agg	−0.6371 ***	−0.5239 ***	−0.9552 ***
	(0.0900)	(0.0570)	(0.0883)
agg2	0.1482 ***	0.1078 ***	0.2451 ***
	(0.0400)	(0.0212)	(0.0348)
Control variables	YES	YES	YES
City Fixed	YES	YES	YES
Time fixed	YES	YES	YES
N	1674	1395	1395

Note: Values in parentheses are standard errors, *** represents significance at the 1% level.

**Table 7 ijerph-20-01575-t007:** Endogenous test results.

Variables	lneil	
	(9)	(10)
	FE	SYS-GMM
L.lneil	-	0.6397 ***
	-	(0.0426)
L.lnec	-	-
	-	-
L.lntc	-	-
	-	-
agg	−0.5394 ***	−0.3300 ***
	(0.1223)	(0.0877)
agg2	0.1110 *	0.0723 **
	(0.0604)	(0.0331)
_cons	−3.1822 ***	0.2163
	(0.7464)	(0.75)
Control variables	YES	YES
City Fixed	YES	-
Time fixed	YES	-
R2	0.515	-
N	4464	4185
AR(1)	-	0.000
AR(2)	-	0.527
Hansen	-	0.191

Note: L.lneil represents the lag of lneil. Values in parentheses are standard errors, * represents significance at the 10% level, ** represents significance at the 5% level, and *** represents significance at the 1% level.

## Data Availability

The data are not publicly available due to issues of personal privacy and non-open access to the research program. The related code is available on request from the corresponding author.

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
