# Peer review of "Research on the Effect of Manufacturing Agglomeration on Green Use Efficiency of Industrial Land"

_ijerph, 2023, doi:10.3390/ijerph20021575_

Round 1

Reviewer 1 Report

The article is very interesting and analyzes the impact of manufacturing agglomeration on green use efficiency of industrial land in China. This is a guideline for the Chinese government to carry out the transformation and upgrading of manufacturing industries and thus enhance the green use efficiency of industrial land. Overall, the article is scientifically relevant to the field of the journal, the logic is clear and the research method is scientific, and its scientific utility is great both for the academic environment and for the authorities. However, there are several problems in the article that need further improvement.

1. The introduction needs to be strengthened.The scientific contribution and novel knowledge of this research need to be specified. What problems should be solved

2.There are some issues that need to be considered when revising the article, such as the inconsistent size of Table 7 text symbols.

3. What were the reasons to choose such variable? Are there any literature positions describing variables? Are there other authors describing such research?

4.References take 25 publications are cited in the entire article. Literature research well started, but not enough publications. It is proposed to add the following articles that contain new research in this area.

5.The discussion or conclusion should refer to research conducted in this field in other countries and cited in this publication. Please complete this and the article will be a valuable scientific contribution.

Author Response

Dear reviewer,

Thank you for your letter and comments concerning our manuscript entitled " Research on the Effect of Manufacturing Agglomeration on Green Use Efficiency of Industrial Land" (ijerph-2137901).

We are grateful to the reviewers for your time and constructive comments on our manuscript. We have implemented your comments and suggestions and wish to submit a revised version of the manuscript for further consideration in the journal. Changes in the initial version of the manuscript are marked up using the “Track Changes” function in the revised version. In order to reflect the suggestions of the reviewers, the revised manuscript is a bit messy, we hope for understand. Below, we also provide a point-by-point response explaining how we have addressed each of the reviewers’ comments. We look forward to the outcome of your assessment.

Yours sincerely,

Yuan Wang

Reviewer 2 Report

This study used panel data of 279 cities and above cities across China from 2004 to 2019 to analyze the spatial and temporal variation characteristics of MA and GUEIL across China and regions, and to investigate the effect of MA on GUEIL empirically. This study provided a reference for understanding the relationship between manufacturing agglomeration and green use efficiency of industrial land, and has important theoretical and practical value. It is suggested to further modify the article from the following aspects.

(1) It is suggested to increase the number of keywords appropriately.

(2) In the research review part, it is suggested to increase the scope of the review, not only from the relationship between the two, but also to increase the necessary summary of the study of their relationship.

(3) The discussion should discuss the main results of the study, not just explain the significance and shortcomings of the study.

(4) It is suggested to check the accuracy of the language and improve the readability of the language.

Author Response

Thank you for your letter and comments concerning our manuscript entitled " Research on the Effect of Manufacturing Agglomeration on Green Use Efficiency of Industrial Land" (ijerph-2137901).

We are grateful to the reviewers for your time and constructive comments on our manuscript. We have implemented your comments and suggestions and wish to submit a revised version of the manuscript for further consideration in the journal. Changes in the initial version of the manuscript are marked up using the “Track Changes” function in the revised version. In order to reflect the suggestions of the reviewers, the revised manuscript is a bit messy, we hope for understand. Below, we also provide a point-by-point response explaining how we have addressed each of the reviewers’ comments. We look forward to the outcome of your assessment.

Yours sincerely,

Yuan Wang
